# Detection of Local Prostate Cancer Recurrence from PET/CT Scans Using Deep Learning

**DOI:** 10.3390/cancers17091575

**Published:** 2025-05-06

**Authors:** Marko Korb, Hülya Efetürk, Tim Jedamzik, Philipp E. Hartrampf, Aleksander Kosmala, Sebastian E. Serfling, Robin Dirk, Kerstin Michalski, Andreas K. Buck, Rudolf A. Werner, Wiebke Schlötelburg, Markus J. Ankenbrand

**Affiliations:** 1Center for Computational and Theoretical Biology, Julius-Maximilians-University Würzburg, 97070 Würzburg, Germany; 2Department of Nuclear Medicine, Dr. Burhan Nalbantoglu State Hospital, Nicosia 99010, Cyprus; huli@hotmail.co.uk; 3Department of Nuclear Medicine, University Hospital Würzburg, 97080 Würzburg, Germanybuck_a@ukw.de (A.K.B.);; 4Department of Nuclear Medicine, LMU University Hospital, LMU Munich, 80539 München, Germany; 5Department of Bioinformatics, Julius-Maximilians-University Würzburg, 97070 Würzburg, Germany

**Keywords:** prostate cancer, PC, PET/CT, positron emission tomography, [^18^F]-PSMA, artificial learning, deep learning, DenseNet

## Abstract

Prostate cancer is a leading cause of cancer-related deaths in men around the world. A type of imaging technique called positron emission tomography (PET), which uses a special scan to detect cancer, has shown great promise in identifying recurring prostate cancer and spread to other parts of the body. In this study, we created a computer-based model that uses PET scan images to predict if prostate cancer has come back after treatment. To improve the model’s performance, we tried different methods, such as focusing on different parts of the image, adding extra information from the patient’s medical history, and including details about whether the patient had prior surgery to remove the prostate. These efforts led to an accuracy of 77% in predicting cancer recurrence. While this accuracy was lower than the desired 90%, the model still showed significant improvement. Many approaches were tested, each helping to improve the model. The results of this study are an important step forward in developing tools that can reliably detect cancer recurrence in the prostate area. However, more research is required to further improve the model’s accuracy and make it more useful for doctors in real-life situations.

## 1. Introduction

Prostate cancer remains one of the leading causes of cancer-related deaths in men, with over 1.2 million new cases diagnosed and more than 350,000 deaths annually [1,2]. Early detection and accurate staging are critical for successful treatment, and therapy effectiveness is typically monitored by tracking serum prostate-specific antigen (PSA) levels. When PSA levels rise, indicating biochemical recurrence, patients are often referred for re-staging, a process that is crucial for determining the most appropriate treatment strategy. A key focus during re-staging is identifying whether there is a local recurrence of cancer in the prostate region. Several diagnostic methods are used for staging and detecting prostate cancer, including digital rectal examination, transrectal ultrasound (TRUS), magnetic resonance imaging (MRI), and positron emission tomography (PET) combined with computed tomography (CT) [3]. Among these, PET/CT has emerged as a reliable technique [4]. In ^18^F-PSMA-PET, the radioactive tracer ^18^F-PSMA-1007 binds selectively to the prostate-specific membrane antigen (PSMA), a protein highly expressed on prostate cancer cells, and it is performed in combination with the CT scan, allowing for precise anatomical imaging. Typically, a whole-body PET/CT scan is conducted to detect potential metastases. Specifically, for prostate cancer diagnosis and staging during remission, each PET/CT examination generates high-dimensional data, comprising hundreds of image slices with thousands of pixels per slice for both PET and CT parts. Interpreting these complex data to answer the critical binary question, i.e., whether there is a local recurrence of prostate cancer in the prostate bed or not, requires trained experts.

Recently, artificial neural networks have shown great potential in assisting with similar tasks, such as tumor segmentation and outcome prediction in head and neck cancers [5], as well as the identification of pathological mediastinal lymph nodes in lung cancer [6]. These advancements highlight the promise of AI in improving diagnostic accuracy and efficiency in prostate cancer staging and treatment [7]. In parallel, the field of theranostics has emerged, promising improvements in therapy as well as diagnostics in oncology [8,9]. Together with the predictive power of novel AI methods, these developments have the potential to dramatically improve early diagnosis, prognostic staging, and patient management.

Recently, deep neural networks have successfully been trained for survival prediction of prostate cancer patients based on magnetic resonance imaging [10] and segmentation and classification of uptake patterns on ^18^F-PSMA-PET scans [11]. Therefore, the aim of this work was to develop a neural network capable of detecting local recurrence in the prostate or prostate bed with a high accuracy of at least 90% using ^18^F-PSMA-1007 PET/CT examinations. This goal of 90% accuracy was somewhat arbitrarily selected as the minimum performance to justify further exploration. For direct clinical application, even higher accuracy is required.

## 2. Materials and Methods

### 2.1. Study Population

Details concerning the study group are shown in Table 1. Briefly, 1145 patients with 1404 ^18^F-PSMA-1007 PET/CT scans performed at the Department for Nuclear Medicine at the University Hospital Würzburg between 2019 and 2023 were included in this study. The PET/CT scans were split into training (1016 scans), validation (188 scans), and test (200 scans) sets.

We used hybrid PET/CT scanners with an extended field-of-view for the PET and a 64- or 128-slice spiral CT (Biograph64 or 128, Siemens Healthineers; Erlangen, Germany). All cases were re-evaluated explicitly for this project by a trained nuclear medicine physician (H.E. using syngo.via, V60A, Siemens Healthineers; Erlangen, Germany) to label each examination with 0 (no local recurrence in the prostate region, n = 637), 1 (local recurrence in the prostate region, n = 704), or 2 (uncertain whether there is local recurrence in the prostate region). Instances with label 2 were excluded from training. In summary, the following metadata were available for each examination in addition to PET/CT scans: patient pseudo-ID, age, sex, staging, prostatectomy status, PSA level, and label.

Even though the question of local recurrence makes sense only at re-staging, we decided to include primary staging patients in the training as they provided examples of scans with cancerous tissue in the prostate region. We assigned all primary staging scans to the training set. The remaining examinations were assigned randomly into training and validation sets, ensuring that all scans from the same patient ended up in the same set.

The test set comprises 200 scans from 198 patients on the same scanners. This set contains re-staging examinations only. However, 83 patients (84 scans) already had scans in the training/validation set, while 115 patients (116 scans) were novel. Even for known patients, these are new examinations, and both metadata and the label might have changed. The parameters for measurement were identical to those from the training/validation set; the same metadata were supplied, and the same labeling procedure was employed.

### 2.2. Data Processing

Images were exported from the clinic PACS in DICOM format and converted to NIfTI via dcm2niix [12]. These NIfTIs contain intensities in Becquerels/milliliter (BQML) in order to convert them to standardized uptake calue (SUV), and scan-specific scaling factors were calculated from DICOM metadata. Data loaders for deep learning in Python (version 3.12.3) were implemented using nibabel (version 5.2.1, [13]), PyTorch (version 2.3.0, [14]), and Monai (version 1.3.1, [15]). CT and PET images for each patient were measured at different resolutions but co-registered to a common space to ensure accurate spatial alignment (Figure 1). Therefore, after reading both files individually, they were re-scaled to a matrix size of 150 × 150 × 150. Depending on the pixel spacing and slice thickness of the original scans, the resulting voxel size differed between patients. Intensity was re-scaled to a range of 0 to 1 based on the minimum and maximum values in each scan. Models with consistent scaling across scans using a fixed linear transformation were explored (Appendix A). Additional transformations like cropping or augmentation transformations were applied depending on the model. Metadata and labels were provided as a simple csv file and loaded via Pandas (version 2.2.2, [16,17]).

### 2.3. Models

As part of this project, many models were trained to explore the effects of including different kinds of data, pre-processing, and deep learning techniques. New models build on previous successful models. In addition to the initial experimentation [18], more than twenty models and variants were trained systematically. Here, we only focus on four models that show incremental improvements. All other models and their variants are described in Appendix A. Source code and trained models are shared in the GitHub repository (version 0.3.0) and on Zenodo. All models use the DenseNet architecture [19] with 121 layers, as implemented in Monai [15]. All models used cross-entropy as the loss function, the Adam optimizer, and a learning rate of 10^−5^ for 15 epochs. The batch size was set to 16, except for Model A, where this exceeded the GPU memory; in this case, a batch size of 8 was used. After each epoch, the validation accuracy was logged, and the weights were saved. We report the maximum accuracy and provide the model weights from the epoch that reached this maximum. Training was performed on an NVIDIA GeForce RTX 4090 GPU with 24 GB of RAM (Nvidia Corporation, Santa Clara CA, USA).

#### 2.3.1. Model A: Base Model

The initial model provided the baseline performance, with the most naïve approach, passing the resized data to a DenseNet121, including moderate augmentation (random rotations by up to 0.2 radians).

#### 2.3.2. Model B: Cropped Around the Prostate

The data include the whole body, while the specific question (local recurrence) can be answered by focusing on the area around the prostate (region). To automatically crop image volumes around the prostate, the location of the prostate was determined by TotalSegmentator (version 2.1.0, [20]). This worked in most cases, even for patients after radical prostatectomy. In the remaining cases, the position of the urinary bladder was used as a proxy (Figure 2). Images were cropped to a 20 cm × 20 cm × 20 cm cube around the centroid of the prostate (or urinary bladder) in patient coordinates and re-scaled to 70 × 70 × 70 voxels. Variants with stronger cropping were explored (Appendix A).

#### 2.3.3. Model C: Adding Prostatectomy Status and PSA Level

The status of prostatectomy (px, 0 = no prostatectomy; 1= radical prostatectomy) and the PSA levels are potentially informative. Therefore, this information was added to the model as separate image layers with repeated values (0 or 1 for px and a floating-point number with the normalized PSA level). Before training model C with integrated px information, we trained separate models for px = 0 and px = 1 (Model 5a, 5b in Appendix A). Those models had high accuracy but low balanced accuracy as they overfit the respective majority class (Appendix A).

#### 2.3.4. Model D: More Extensive Augmentation and Hyperparameter Optimization

Training usually converged rather quickly. So, to increase data heterogeneity (without generating new training sets), we employed more extensive augmentation. With a growing number of augmentation transformations, the number of hyperparameters to tune also increased. Therefore, we applied the hyperparameter optimization toolkit Optuna (version 3.6.0) [21] to determine optimal hyperparameter combinations for spatial cropping around the center (CenterSpatialCropd), random flipping (RandFlipd), and random zooming (RandZoomd).

### 2.4. Evaluation

The classification performance of all models was evaluated using the accuracy and the balanced accuracy of the validation set. Accuracy is the number of correct predictions divided by the number of total predictions. This metric is susceptible to class imbalance [22]. Therefore, this metric is complemented by the balanced accuracy, which is equivalent to the arithmetic mean of sensitivity and specificity in the case of binary classification.

Only the best-performing model was evaluated on the hold-out test set. Missing values were imputed before prediction in the few cases with missing metadata (two cases missing px; eleven cases missing PSA). For px, the majority class (px = 1), and PSA, mean values (30.1) were used. In addition to accuracy and balanced accuracy, the confusion matrix is reported. Furthermore, accuracy is separately determined for known patients and novel patients.

## 3. Results

The initial model A, which used the entire volume with moderate augmentation, reached an accuracy of 56.4% with a balanced accuracy of 49.5% but an F1-score of 0.0, indicating a complete lack of true positive predictions (Table 2, Figure 3). Restricting the input data to a 70 × 70 × 70 volume around the prostate gland, model B reached an accuracy of 70.7% and a balanced accuracy of 67.9%. Including prostatectomy status and PSA levels as additional data layers (model C) further increased the accuracy to 76.5% with a balanced accuracy of 73.8%. Finally, using Optuna to optimize the hyperparameters and augmentation settings returned the following settings: CenterSpatialCropd with dimensions 65 × 46 × 69, RandFlipd along the spatial axis 1 with a probability of 1 (100%), and RandZoomd with a minimum and maximum zoom of 0.5 and a probability of 1 (100%). Model D trained with these settings had the highest accuracy (77.1%) and balanced accuracy (73.9%). Neither of these models came close to our target accuracy of 90%. As the model with the highest validation accuracy (Table 2), we selected model D for evaluation on the hold-out test set.

Model D reached an accuracy of 71.0% and a balanced accuracy of 72.8% on the test set (Figure 3). While the model had high specificity (88.8%), the sensitivity was poor (56.8%). In 116 cases, the patients were not previously known. Of these cases, 74.1% were correctly identified, while 66.7% of the 84 cases with previously known patients were correctly identified.

## 4. Discussion

This work aimed to develop a highly accurate neural network to predict local recurrence in prostate cancer patients from PSMA-PET/CT images. We iteratively refined our approach as the initial naïve model had insufficient accuracy with no true positives. Specifically, we cropped the images around the prostate gland to focus on this region, included important metadata as separate image layers, and performed hyperparameter optimization of certain augmentations to increase the data heterogeneity (Table 1). Furthermore, we modified the dimensions of the region of interest and attempted to mask the PET channel in certain organs or everywhere except the prostate to let the model focus on the prostate alone (Appendix A). Interestingly, some refinements, like using a consistent intensity scaling based on SUV units across all patients (Section A.2.11), decreased the classification accuracy rather than improving it. This indicates that our models were unable to learn the expected proper features from the data. While most of our refinements generally increased the accuracy, the final test set accuracy of 71.0% is still far from acceptable in a clinical context. In particular, the poor sensitivity is concerning in this context as it corresponds to patients with local recurrence not being identified. Sensitivity can generally be increased using a weighted loss function or adapting the threshold for the predicted probability. However, this usually trades specificity for sensitivity. As the overall performance of our models was far from ideal, we did not attempt fine-tuning this trade-off. Furthermore, attempts to use Grad-CAM [23] to determine which regions within the volume of interest contribute most to the decision of the model returned ambiguous results. This indicates that the features that were learned are not the medically relevant features of the image. While we cannot postulate that we reached a theoretical accuracy limit for this task (given our data set), we employed many advanced techniques to improve our models iteratively and reached a comparable accuracy to the 71.4% previously reported for the prediction of prostate cancer recurrence based on ^18^F-FACBC PET [24]. By documenting this evolution and publishing all code and models, we provide a blueprint for approaching DL to answer a medical research question. While it is undoubtedly possible to marginally increase the accuracy of artificial neural networks by extending the hyperparameter space (e.g., by exploring different network architectures), we postulate that to reach accuracies above 90%, a completely different technique, sophisticated domain-specific adaptations (e.g., improved attenuation correction [25,26,27,28]), or much more data are required. Transfer learning is an interesting technique that lowers the necessary training data [29,30]. A pre-trained model on a related task is required to perform transfer learning. Alternatively, a large dataset for a related task can be used to pre-train a model yourself. Even though transfer learning has already been successfully applied for prostate cancer detection on biopsy images [31] and ultrasound [32], we could not find any suitable model or dataset, but now we provide our models as starting points for transfer learning by others. As datasets can usually not be shared because of data protection requirements, we hope our models will be used for TL and that the resulting models will be shared again. We plan to evaluate the performance of an iteratively re-trained model on the initial dataset.

## 5. Conclusions

For the presented task, 1404 examinations were insufficient to reach an accuracy of over 90% even when employing data augmentation, including additional metadata and performing automated hyperparameter optimization. The low F1-score and AUC values indicate that none of the presented models produce reliable results. However, we will facilitate future research and the development of better models by openly sharing our source code and all pre-trained models for transfer learning.

## Figures and Tables

**Figure 1 cancers-17-01575-f001:**
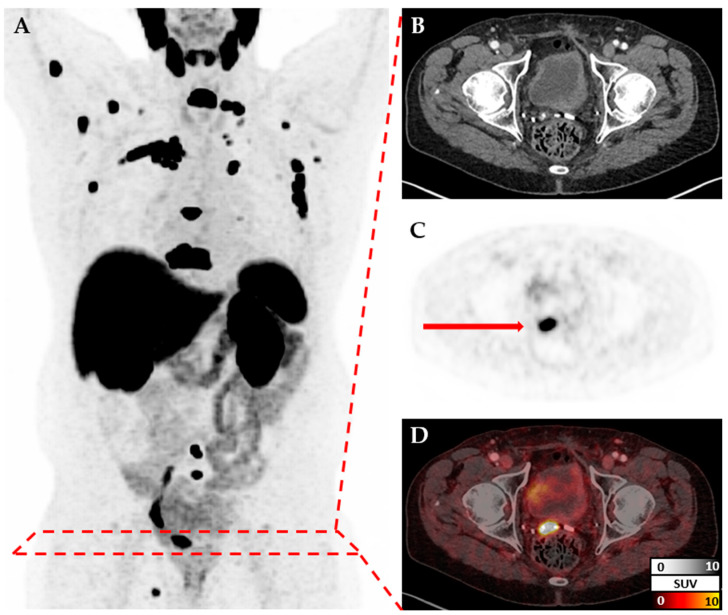
PSMA-PET/CT: (**A**) maximum intensity projection; axial corresponding CT (**B**) and PET (**C**) slices; (**D**) fused PET and CT slices. Example of a 78-year-old patient with osseus and lymphnodal metastases, as well as a local recurrence in the prostate bed (indicated by the red arrow).

**Figure 2 cancers-17-01575-f002:**
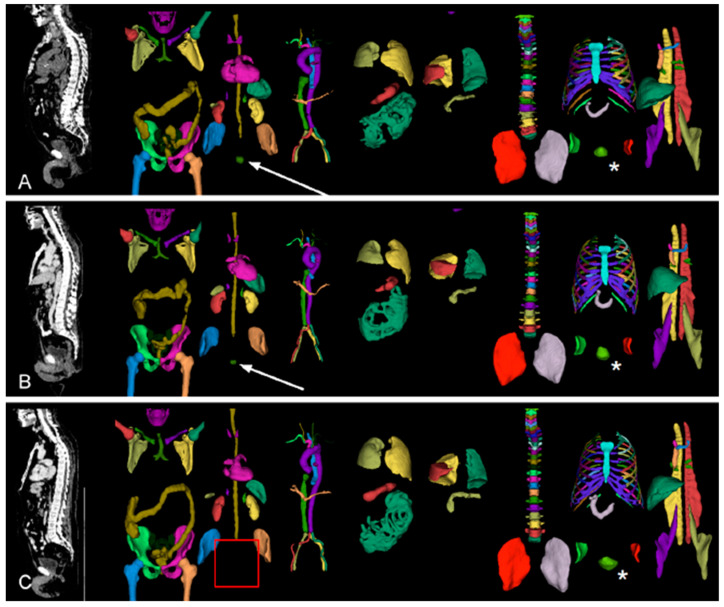
Result of TotalSegmentator [20] on three patients in Subfigures (**A**), (**B**), and (**C**). White arrows indicate the detected prostate gland in patients A and B, while the red rectangle indicates the absence of the prostate gland in patient C. The white asterisks (*) indicate that the urinary bladder was successfully detected in all patients. Color indicates the organ that was detected by TotalSegmentator.

**Figure 3 cancers-17-01575-f003:**
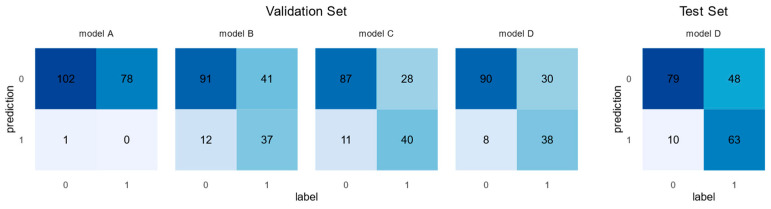
Confusion matrices of models A–D on the validation set (**left**) and model D on the test set (**right**). Columns are labels, and rows are predictions. The numbers within the cells are additionally color coded with a gradient from light blue (small numbers) to dark blue (large numbers).

**Table 1 cancers-17-01575-t001:** Study group characteristics.

	Total	Training	Validation	Test
Patient number	1145	868	161	198
^18^F-PSMA-PET/CT scan number	1404	1016	188	200
**Indication for ^18^F-PSMA-PET/CT scan**				
Primary staging	212	212	0	0
Re-staging	1192	804	188	200
**Patients’ characteristics**				
Age, mean (range)	70.5 (44–90)	70.3 (44–90)	70.9 (46–89)	71.3 (53–86)
Scans with prior prostatectomy (%)	796 (57%)	546 (54%)	127 (68%)	123 (62%)
PSA-level, mean (range)	44.9 (0–7434)	46.3 (0–3420)	53.8 (0–7434)	30.1 (0–932)
**Label**				
0 (no local recurrence)	637	445	103	89
1 (local recurrence)	704	515	78	111
2 (uncertain case)	63	56	7	0

**Table 2 cancers-17-01575-t002:** Model accuracies, balanced accuracies, F1-scores, and area under the receiver operating characteristic curve (AUC) on the validation set.

Model	Accuracy	Balanced Accuracy	F1-Score	AUC
Model A: Base Model	0.564	0.495	0.000	0.500
Model B: Cropped FOV	0.707	0.679	0.583	0.697
Model C: px and PSA	0.765	0.738	0.672	0.736
Model D: hyperparam.	0.771	0.739	0.667	0.753

## Data Availability

The datasets presented in this article are not readily available because of data protection regulations for routine clinical data. Requests to access the datasets should be directed to WS. All source code is available at https://github.com/BioMeDS/f18-psma-pet-ct-ai (accessed on 2 May 2025) and archived at Zenodo https://doi.org/10.5281/zenodo.14944344. The pre-trained model weights are deposited on Zenodo https://doi.org/10.5281/zenodo.14944879.

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
