# Peer review of "Detection of Local Prostate Cancer Recurrence from PET/CT Scans Using Deep Learning"

_cancers, 2025, doi:10.3390/cancers17091575_

Round 1
Reviewer 1 Report
Comments and Suggestions for Authors Overall, a good, original and well-founded idea. The implementation of the set goal required several adjustments and the addition of several steps (additional programs to further balance the still weak results). In the ​​materials and methods, the numbers are not presented in an immediately understandable way; it requires reading through several times until a clearer structure can be recognized. This is also the core weakness of the paper: the planned number of patients or data sets for this experiment is too small. This is also clearly described in the conclusions, but this is a foreseeable weakness in the study design when it comes to training AI. Admittedly, the required number of data sets for AI is difficult to estimate. The disclosure of the scripts and programming (in the appendix) for further use must therefore be credited positively.Author Response
Response to Reviewer 1
Overall, a good, original and well-founded idea. The implementation of the set goal required several adjustments and the addition of several steps (additional programs to further balance the still weak results).
Thank you for your positive evaluation of our work.
In the materials and methods, the numbers are not presented in an immediately understandable way; it requires reading through several times until a clearer structure can be recognized.
Thank you for the valuable feedback. We rephrased the sentences with numbers in the materials and methods section. Further, we no longer report the cases that were later excluded because of failed conversions (updating all numbers, including Table 1). This simplifies the description of the data set.
This is also the core weakness of the paper: the planned number of patients or data sets for this experiment is too small. This is also clearly described in the conclusions, but this is a foreseeable weakness in the study design when it comes to training AI. Admittedly, the required number of data sets for AI is difficult to estimate. The disclosure of the scripts and programming (in the appendix) for further use must therefore be credited positively.
We agree that the dataset size is the major limitation of our study. As we can not simply scale up this number, we hope that readers still find our detailed workflow description, scripts and models useful. We are happy that you identified these points as valuable.
Reviewer 2 Report
Comments and Suggestions for Authors
The authors developed multiple deep learning models to detect Prostate cancer recurrence using PET CT images. They attempted to improve performance using different approaches and reported the results. This kind of comprehensive studies are a very beneficial and valuable contribution to the field as they pave the way for further studies and can be a starting point for other researchers and teams. However, the study results may not be reliable as there are some basic problems in the methodology and results presented as well as the final conclusion. The study needs to be revised by clinicians, especially medical physicists and nuclear medicine MDs. The detailed comments can be found below.
- Line 97: dcm2niix converts the PET images to BQML unit. Nowhere in the manuscript mentioned converting them to SUV unit. This is a serious problem and limitation when handling PET images. Consultation with a clinical partner including a medical physicist and an MD is highly recommended.
- Line 100: PET and CT images do not cover the same volume. It is better to say they are co-registered. Better to mention voxel spacing as well in addition to matrix size of 150×150×150. If the voxel spacing was variable among images they would cover different areas. Handling the images with matrix size instead of voxel spacing limits generalizability.
- Line 117: please update weights to trained models for clarity.
- Model B: did you check the crop visually? Why not select a bone landmark such as hip or sacrum with a better segmentation accuracy compared to bladder or prostate?
- Figure 2: you used urinary bladder and or prostate segmentation? What is the point in visualizing all the organs? Caption of Figure 2 says successfully detected bladder in all cases? Did you check them all visually? Any corrections? How did you ensure the lesion is in the cropped area? We assume that the classification model is seeing the lesion and locoregional invasion when classify the images.
- Model C: this approach is not adding PX to the model. This is training a specific model for each group. Limiting the data balance in each group. Table A2 needs to be transferred to the manuscript’s body for clarification.
- Model D: randzoom. We expect the model to see the lesion somewhere in the field of view and decide about locoregional status. the randzoom function may exclude the lesion while the label is 1.
- Evaluations: F1 score need to be reported. ROC curve is needed too. GradCam and saliency maps are recommended.
- The reason behind selecting 90% as a goal is not clear. Why not 80 or 95? What is clinical relevance?
- Please plot the confusion matrix instead of showing them as a table. And not normalized confusion matrix. For all the training strategies and not only for one strategy.
- Model 4.b. the task is about locoregional invasion. You masked the image to prostate, even with a 100% accuracy model, this is not explainable. Not clinically relevant to the task.
- Supplementary file section2. Methods and results. ?? in the text. Same as 2.1.
- AUC is an accepted and well-known terminology no need to mention it as AUROC. Dicom and NIFTI are well known, no need to mention the full name.
- Intra/inter observers may be addressed by adding two reviewers for labeling the data. Please mention the software used by physicians for labelling the data. 2.1 can be summarized with using the term blind review.
- Train transformation raises a lot of questions. ScaleIntensityd function from monai.transforms normalizes the image to its own maximum and minimum. The CT may be from -2000 to +4000 in images with a metal artifact, the voxel intensity in another CT may be from -1000 to +1500, this kind of normalization is not appropriate at all and question the whole approach. Same for a PET image in BQML format. BQML is affected by patient size, injected activity and time from injection to scan, that is the reason behind definition of SUV. The whole study needs to be repeated with strong communication with a clinician in nuclear medicine. RandRotated function copied from a segmentation Monai example where the input is an image and output are a segmentation. Why is the CT interpolated using bilinear and PET using nearest neighborhood?
- The whole-body image dimensions are usually around 1.5 meter in Z axis and around 50 0r 60 cm in axial plane, what is the justification in resizing the image to 150 150 150 dimensions regardless of patient height, scan range and FOV?
- Lots of additional unnecessary details about what EnsureChannelFirstd function in MONAI which is not even necessary and deprecated in the mentioned version and it is included as a default true value in LoadImaged function.
- Please use coherent font in all the text. (e.g., table A.4)
- Figure A1 and figure 2 are the same!
- crop_pet_ct_by_pro function assumes the images are always in the same voxel spacing. Need to be updated for generalizability in other scanners and voxel spacings.
- Figure A3. The uptake in the left hip bone is a pathologic uptake! You masked it then!
- Line 220: “While it is undoubtedly possible to marginally increase the accuracy of artificial neural networks by extending the hyperparameter space (e.g., by exploring different network architectures), we postulate that to reach accuracies above 90%, a completely different technique, sophisticated domain-specific adaptations (e.g. attenuation correction[19]) or much more data is required. Transfer learning is an interesting technique that lowers the necessary training data. [20].” Does it mean that you included non-attenuation and scatter corrected images in your study? Figure 1 is clearly a corrected image.
- In the training code: complete_loader uses the data from complete_ds from complete_data and reads it from df which has all the data. Prediction was done on all data? It is not clear if it is a data leakage or not.
Author Response
Response to Reviewer 2
The authors developed multiple deep learning models to detect Prostate cancer recurrence using PET CT images. They attempted to improve performance using different approaches and reported the results. This kind of comprehensive studies are a very beneficial and valuable contribution to the field as they pave the way for further studies and can be a starting point for other researchers and teams. However, the study results may not be reliable as there are some basic problems in the methodology and results presented as well as the final conclusion. The study needs to be revised by clinicians, especially medical physicists and nuclear medicine MDs. The detailed comments can be found below.
Thank you for your critical evaluation of our work. Please find our response to the excellent points you raised, below.
- Line 97: dcm2niix converts the PET images to BQML unit. Nowhere in the manuscript mentioned converting them to SUV unit. This is a serious problem and limitation when handling PET images. Consultation with a clinical partner including a medical physicist and an MD is highly recommended.
Thank you for bringing this important point to our attention. The PET files were indeed used with BQML units. The way we previously did our training, this did not matter as each patient was individually normalized to a range of 0 to 1 and the conversion to SUV is done by multiplying a scaling factor. However, as you noted in point 15 below, this approach of normalizing each patient is problematic. Therefore, we did convert the cropped images to SUV units by calculating the scaling factors from the Dicom metadata. These images were used to re-train models with updated transformations (see our response to point 15 below).
- Line 100: PET and CT images do not cover the same volume. It is better to say they are co-registered. Better to mention voxel spacing as well in addition to matrix size of 150×150×150. If the voxel spacing was variable among images they would cover different areas. Handling the images with matrix size instead of voxel spacing limits generalizability.
You are right. We updated our wording and explicitly state the limitation of different voxel sizes between patients in the material and methods section.
- Line 117: please update weights to trained models for clarity.
Done
- Model B: did you check the crop visually? Why not select a bone landmark such as hip or sacrum with a better segmentation accuracy compared to bladder or prostate?
The cropped regions were visually inspected, and no discrepancies were observed. The bladder was chosen as a reference structure due to its reliable identification in the TotalSegmentator model.
- Figure 2: you used urinary bladder and or prostate segmentation? What is the point in visualizing all the organs? Caption of Figure 2 says successfully detected bladder in all cases? Did you check them all visually? Any corrections? How did you ensure the lesion is in the cropped area? We assume that the classification model is seeing the lesion and locoregional invasion when classify the images.
The statement in Figure 2's caption refers to the successful detection of the prostate gland in the presented cases. To ensure the lesion was included in the cropped area, we verified that the region of interest encompassed both the lesion and potential locoregional invasion, aligning with the assumption that the classification model is focusing on these areas during image analysis.
- Model C: this approach is not adding PX to the model. This is training a specific model for each group. Limiting the data balance in each group. Table A2 needs to be transferred to the manuscript’s body for clarification.
Sorry for the confusion. Model C corresponds to Model 6c in the appendix which does contain px and psa information encoded in separate image layers. We explored separate models for each group with Model 5a and 5b but discarded them, because of the class imbalance in each group. Therefore, Model 6c builds on top of Model 3 and is trained on the full cohort. We re-wrote the section in material and methods (2.3.3) to make it clearer.
- Model D: randzoom. We expect the model to see the lesion somewhere in the field of view and decide about locoregional status. the randzoom function may exclude the lesion while the label is 1.
That's right. A too large zoom factor might zoom in to a region that does not contain the lesion. We allowed Optuna to optimize the minimum and maximum zoom factor within a range of 0.5 and 1.5 (values below 1 are zooming out so the field of view remains unrestricted, values above 1 are zooming in and thereby limiting the fov, we assume that with a zoom factor of 1.5 the lesion (expected to be centered within the image) remains within view.
- Evaluations: F1 score need to be reported. ROC curve is needed too. GradCam and saliency maps are recommended.
We added F1 score and AUC to Table 2 in the main text. We did experiment with GradCam during model development, but in our 3D case we had a hard time properly visualizing and interpreting the results. In our cases there was no clear focus on the prostate bed which indicates that the model is not using the desired features. This also fits the observation that the models are not performing very well, so we don’t expect them to have learned how to correctly identify local recurrence.
- The reason behind selecting 90% as a goal is not clear. Why not 80 or 95? What is clinical relevance?
Agreed, this goal is arbitrary. It was selected after initial discussions as the minimum performance required to justify further exploration. For clinical application a much higher accuracy would be required. We now explain this choice in the introduction.
- Please plot the confusion matrix instead of showing them as a table. And not normalized confusion matrix. For all the training strategies and not only for one strategy.
We replaced Table 3 with Figure 3 showing the confusion matrix on the validation set for all four models and the confusion matrix on the test set for model D.
- Model 4.b. the task is about locoregional invasion. You masked the image to prostate, even with a 100% accuracy model, this is not explainable. Not clinically relevant to the task.
Sorry for the confusion. While in model 4a we masked regions of UB, hip bone, etc. In model 4b we masked everything except for the prostate (bed). So only signal from the prostate region was retained while all other signal was removed. We added a clarifying sentence to section 2.7.2 in the appendix.
- Supplementary file section2. Methods and results. ?? in the text. Same as 2.1.
Thank you. We fixed the broken references.
- AUC is an accepted and well-known terminology no need to mention it as AUROC. Dicom and NIFTI are well known, no need to mention the full name.
We renamed AUROC to AUC and removed the long versions for Dicom and NIFTI.
- Intra/inter observers may be addressed by adding two reviewers for labeling the data. Please mention the software used by physicians for labelling the data. 2.1 can be summarized with using the term blind review.
We agree that inter observer variability would be interesting to know. Sadly, we do not have the capacity to independently re-label all 1404 cases. We added the software used during the labelling procedure to the material and methods section (syngo.via, V60A, Siemens Healthineers; Erlangen, Germany).
- Train transformation raises a lot of questions. ScaleIntensityd function from monai.transforms normalizes the image to its own maximum and minimum. The CT may be from -2000 to +4000 in images with a metal artifact, the voxel intensity in another CT may be from -1000 to +1500, this kind of normalization is not appropriate at all and question the whole approach. Same for a PET image in BQML format. BQML is affected by patient size, injected activity and time from injection to scan, that is the reason behind definition of SUV. The whole study needs to be repeated with strong communication with a clinician in nuclear medicine.
This is indeed an excellent point, thanks for bringing this up. One of the most counter-intuitive observations we made was that scaling images independently (using ScaleIntensityd) led to higher accuracy than consistent scaling (e.g. using ScaleIntensityRanged). From a human perspective, this is unexpected as voxel values are no longer independently interpretable. Images without any bright spots seem to have high intensity everywhere after scaling, because the background noise gets amplified by the scaling. While this is not what humans are used to, the information about the absolute intensities is still encoded to some degree in the amount the background signal gets amplified. Apparently, it is easier for our deep learning model to learn from this data, than from the scaled data where the background is all the same but absolute values in bright regions carry the meaning. To demonstrate this, we re-trained models B, C, and D with ScaleIntensityRanged with a range of -1024 to 3071 for CT and a range of 0 to 106 for the SUV value mapped to the range 0 to 1 (without clipping). With otherwise unchanged hyperparameters the accuracy dropped for Model B from to, for Model C from to, and for Model D from to. We still think, that with more training data, to learn more meaningful features, these models might outperform the others. Therefore, we include the code of these variants in GitHub and the trained models in Zenodo. We added discussion of this phenomenon to the paper and a section to the appendix.
RandRotated function copied from a segmentation Monai example where the input is an image and output are a segmentation. Why is the CT interpolated using bilinear and PET using nearest neighborhood?
Thanks for catching this. It indeed does not make sense and bilinear is used for both PET and CT in the re-trained models.
- The whole-body image dimensions are usually around 1.5 meter in Z axis and around 50 0r 60 cm in axial plane, what is the justification in resizing the image to 150 150 150 dimensions regardless of patient height, scan range and FOV?
This is only done in the initial naïve model that was supposed to serve as a baseline with as little domain knowledge included as possible. In all other models a smaller and cubic region around the prostate (or UB) were used.
- Lots of additional unnecessary details about what EnsureChannelFirstd function in MONAI which is not even necessary and deprecated in the mentioned version and it is included as a default true value in LoadImaged function.
Thank you very much. We removed EnsureChannelFirstd from our code and from the text in the appendix.
- Please use coherent font in all the text. (e.g., table A.4)
We had Model 1 as baseline model in italics. We agree that this did not improve clarity so we removed the formatting.
- Figure A1 and figure 2 are the same!
Thanks, we removed the figure from the appendix and referenced figure 2 there.
- crop_pet_ct_by_pro function assumes the images are always in the same voxel spacing. Need to be updated for generalizability in other scanners and voxel spacings.
Sorry for the confusion. The function internally calls from_ct_to_patient and from_patient_to_pix to convert coordinates from voxel to patient space for both ct and pet based on the affine matrices in the niftis. As these functions were previously not shown in Codeblock A5 it was impossible to see. We now added the code of these functions to the code block. For the full implementation, see https://github.com/BioMeDS/f18-psma-pet-ct-ai/blob/main/code/preprocessing/crop_by_prostate_or_ub.py
- Figure A3. The uptake in the left hip bone is a pathologic uptake! You masked it then!
That’s right. While it is certainly undesirable in a real clinical setting to hide metastases, for the question of local recurrence only, this signal might be irrelevant. This is specifically why we tested masking of neighboring tissues to avoid confusion between metastases and local recurrence. However, as this approach did not improve accuracy, we abandoned it anyway.
- Line 220: “While it is undoubtedly possible to marginally increase the accuracy of artificial neural networks by extending the hyperparameter space (e.g., by exploring different network architectures), we postulate that to reach accuracies above 90%, a completely different technique, sophisticated domain-specific adaptations (e.g. attenuation correction[19]) or much more data is required. Transfer learning is an interesting technique that lowers the necessary training data. [20].” Does it mean that you included non-attenuation and scatter corrected images in your study? Figure 1 is clearly a corrected image.
Sorry for the confusion. Our images are indeed attenuation and scatter corrected. We wanted to mention improved attenuation correction methods that work better than current standard methods in the presence of potential problems like motion or other CT artifacts. We rephrased the statement and added additional references to such approaches.
- In the training code: complete_loader uses the data from complete_ds from complete_data and reads it from df which has all the data. Prediction was done on all data? It is not clear if it is a data leakage or not.
We did not use the complete_loader during training or validation. It was only used for re-running the final model on all data to get predictions for all cases. Furthermore, this loader did not contain the test set which is completely independently stored in its own table. In order to avoid confusion, we removed the complete loader from the notebooks of models we re-ran as part of this revision.
Reviewer 3 Report
Comments and Suggestions for Authors
The title “Detection of local prostate cancer recurrence from PET/CT scans using deep learning” seems appropriate for the topic covered.
The manuscript discusses the development of a neural network capable of detecting local recurrence in the prostate or prostatic bed with a high accuracy of at least 90% using 18F-PSMA-1007 PET/CT scans.
The topic is very interesting and current, but the structure of the manuscript needs to be improved. Some sections need to be significantly expanded, others modified in response to comments.
Please see the comments below.
- Authors are strongly encouraged to consider the following state of the art:
-
- The authors are strongly encouraged to expand the introduction and discussions to include theranostics in oncology. This is done through the use of nanosystems that help the process of diagnosis but also therapy in molecular imaging with medical radioisotopes. [10.3390/life14060751], [10.3390/cancers16193323] in these, the authors discuss the potential of these recent strategies to improve the clinical outcomes of patients, optimizing early diagnosis and management of different types of difficult-to-manage cancer such as prostate cancer and the importance of personalized medicine and advances in targeted cancer therapies, explore potential methods to improve prognosis and treatment of cancer in the future from in vitro to patient. This addition at the background or discussion level, would enhance the impact of the interesting work presented. It would bridge the discussed predictive side where AI plays a leading role, to the very hot part related to therapies. The clinical picture would be more complete and the readership larger.
- [10.1038/s41698-023-00481-x][10.1007/s10278-024-01104-y] Here the authors developed an automatic AI-based framework for segmentation and classification of uptake patterns into suspicious and non-suspicious foci for cancer from whole-body DCFPyL [18F] PET/CT images of patients with biochemically recurrent and/or metastatic prostate cancer; furthermore, the development of deep learning models for lesion characterization and outcome prediction in prostate cancer patients using prostate-specific membrane antigen (PSMA) PET/CT imaging has been well described. It is important that these works be considered for final discussions, as they have a great deal of influence on performance comparisons between what is new and what has already been accomplished.
- A list of abbreviations at the end of the manuscript should be included.
- In table 1 "18F-PSMA-PET/CT scan number" seems to be written in a different format. Please correct.
- English is fine.
Finally, it would be helpful to extend the references to enhance the coherence of the article.
Author Response
Response to Reviewer 3
The title “Detection of local prostate cancer recurrence from PET/CT scans using deep learning” seems appropriate for the topic covered. The manuscript discusses the development of a neural network capable of detecting local recurrence in the prostate or prostatic bed with a high accuracy of at least 90% using 18F-PSMA-1007 PET/CT scans. The topic is very interesting and current, but the structure of the manuscript needs to be improved. Some sections need to be significantly expanded, others modified in response to comments.
Thank you for your positive evaluation of our work.
Please see the comments below.
- Authors are strongly encouraged to consider the following state of the art:
- The authors are strongly encouraged to expand the introduction and discussions to include theranostics in oncology. This is done through the use of nanosystems that help the process of diagnosis but also therapy in molecular imaging with medical radioisotopes. [10.3390/life14060751], [10.3390/cancers16193323] in these, the authors discuss the potential of these recent strategies to improve the clinical outcomes of patients, optimizing early diagnosis and management of different types of difficult-to-manage cancer such as prostate cancer and the importance of personalized medicine and advances in targeted cancer therapies, explore potential methods to improve prognosis and treatment of cancer in the future from in vitro to patient. This addition at the background or discussion level, would enhance the impact of the interesting work presented. It would bridge the discussed predictive side where AI plays a leading role, to the very hot part related to therapies. The clinical picture would be more complete and the readership larger.
- [10.1038/s41698-023-00481-x][10.1007/s10278-024-01104-y] Here the authors developed an automatic AI-based framework for segmentation and classification of uptake patterns into suspicious and non-suspicious foci for cancer from whole-body DCFPyL [18F] PET/CT images of patients with biochemically recurrent and/or metastatic prostate cancer; furthermore, the development of deep learning models for lesion characterization and outcome prediction in prostate cancer patients using prostate-specific membrane antigen (PSMA) PET/CT imaging has been well described. It is important that these works be considered for final discussions, as they have a great deal of influence on performance comparisons between what is new and what has already been accomplished.
Thank you for these great suggestions. We expanded both the introduction and discussion sections accordingly.
- A list of abbreviations at the end of the manuscript should be included.
Based on the author instructions of the journal and the template, there is no list of abbreviations (only a long form with short form in parentheses at the first use). We do have a list of abbreviations in the appendix.
- In table 1 "18F-PSMA-PET/CT scan number" seems to be written in a different format. Please correct.
Thank you for the comment. We have changed the words to the correct font format.
- English is fine.
Thank you
- Finally, it would be helpful to extend the references to enhance the coherence of the article.
Thank you, we added multiple additional references to the introduction and discussion.
Round 2
Reviewer 2 Report
Comments and Suggestions for Authors
Thank you so much for updating the manuscript. the text is improved, especially the discussion and introduction are more clear now. They retrained some of the models with wide range of SUV normalization and CT normalization and reported the inferior results compared to what have been reported in the initial submited paper. they updated the Zenodo files and now provided the updated models.
However, there are few limitations for this study which got more serious with the updated confusion matrix and AUC and F1 score values reported.
especially when GradCam results do not support the decision making requirement, the approppriate traiing may be questionble.
Author Response
Comments 1: Thank you so much for updating the manuscript. the text is improved, especially the discussion and introduction are more clear now. They retrained some of the models with wide range of SUV normalization and CT normalization and reported the inferior results compared to what have been reported in the initial submited paper. they updated the Zenodo files and now provided the updated models.
Response 1: Thank you very much for acknowledging the changes and for your comments that led to these improvements.
Comments 2: However, there are few limitations for this study which got more serious with the updated confusion matrix and AUC and F1 score values reported. especially when GradCam results do not support the decision making requirement, the approppriate traiing may be questionble.
Response 2: Thanks for pointing out the remaining limitations. We agree that the confusion matrices, AUC, F1 scores, and failed attempts to use GradCam all underline the overall poor performance of our models. Hence, we conclude that the model has not yet learned to make decisions based on medically meaningful features. We mention GradCam now in the discussion and formulate the limited reliability of all our models in the conclusion.
- line 251: “Furthermore, attempts to use Grad-CAM [23] to determine which regions within the volume of interest contribute most to the decision of the model returned ambiguous results. This indicates that the features that were learned are not the medically relevant features of the image.”
- line 277: “The low F1 and AUC values indicate that none of the presented models produce reliable results.”